# Reproductive Biology and Rearing Improvements of *Apanteles opuntiarum*, Potential Biocontrol Agent of the Argentine Cactus Moth, *Cactoblastis cactorum*

**DOI:** 10.3390/insects15080604

**Published:** 2024-08-10

**Authors:** Laura Varone, Nicole Benda, Mariel Eugenia Guala, Juan José Martínez, Octavio Augusto Bruzzone

**Affiliations:** 1Fundación para el Estudio de Especies Invasivas, Hurlingham 1686, Buenos Aires, Argentina; marielguala@fuedei.org; 2Consejo Nacional de Investigaciones Científicas y Técnicas, Ciudad Autónoma de Buenos Aires 1033, Argentina; asaphes@gmail.com (J.J.M.);; 3Division of Plant Industry, Florida Department of Agriculture and Consumer Services, Gainesville, FL 32608, USA; nicole.benda@fdacs.gov; 4Departamento de Ciencias Biológicas, Facultad de Ciencias Exactas y Naturales, Universidad Nacional de La Pampa, Santa Rosa 7263, La Pampa, Argentina; 5Instituto de Investigaciones Forestales y Agropecuarias Bariloche, Bariloche 8400, Río Negro, Argentina

**Keywords:** parasitism, immature stages, host exposure

## Abstract

**Simple Summary:**

The cactus moth is from South America and feeds on prickly pear cacti (*Opuntia* sp.). However, it is now present in North America and threatening the native *Opuntia* plants there. We are investigating a potential biological control agent, the wasp *Apanteles opuntiarum*, as a sustainable method to control the cactus moth population. This wasp is a larval parasitoid. We studied several aspects of *A. opuntiarum* reproduction and laboratory rearing. We documented the morphology of the wasp eggs and larvae. We found that *A. opuntiarum* prefers to lay its eggs on previously parasitized larvae compared to unparasitized larvae. We also found that exposing cactus moth larvae without the cactus and in groups of 20 larvae resulted in the highest parasitism and produced the most wasp offspring, suggesting that these factors reduced defensive behaviors of the cactus moth larvae. Understanding the wasp’s preference for previously parasitized larvae and how the cactus moth larvae interact with their environment will allow us to manipulate these factors to improve wasp production in laboratory colonies. Efficient laboratory production will be critical for successful release of this wasp as a biological control for the invasive cactus moth.

**Abstract:**

The cactus moth, *Cactoblastis cactorum* (Berg) (Lepidoptera: Pyralidae) is native to South America and has been used as a biocontrol agent of *Opuntia* (Cactaceae) in Australia and South Africa. Its invasion in North America has raised concerns for the native *Opuntia* in the USA and Mexico. We investigated the reproductive biology and rearing procedures of a host-specific potential biocontrol agent, *Apanteles opuntiarum* Martínez and Berta (Hymenoptera: Braconidae). Given the gregarious nature of the parasitoid larvae, we studied the morphology of the immature stages and evaluated evidence of polyembryony and superparasitism. We also investigated the effects of host exposure arena and host density on parasitism rates and wasp production. The morphological descriptions provide a basis for comparison with other species. Early larval instars of *A. opuntiarum* are similar to other microgastrine immature stages. However, the mature larva exhibits placoid sensilla in the epistomal region, a unique character not previously reported. We provide evidence that *A. opuntiarum* eggs are not polyembryonic; females frequently superparasitize and have an oviposition preference for larvae parasitized 1–2 d previously. Exposing larvae of *C. cactorum* to wasps while inside the cactus resulted in lower parasitism and fewer offspring from each host than exposing larvae in the arena without the cactus. Parasitism and mortality rates were higher at lower host densities, possibly due to reduced host group defensive behavior. These results suggest that preference for superparasitism, host defensive behavior, and interactions with the cactus may play an important role in per-host wasp production under laboratory conditions.

## 1. Introduction

The cactus moth *Cactoblastis cactorum* (Berg) (Lepidoptera: Pyralidae) is native to South America and feeds on *Opuntia* cacti. Its larvae are gregarious miners that hollow out and destroy the cladodes. Due to its voracity and host specificity, *C. cactorum* was introduced from Argentina to Australia in 1926 and South Africa in 1933 as a successful biological control agent to manage exotic *Opuntia* species [1,2,3], together with *Dactylopius opuntiae* (Cockerell) (Hemiptera: Dactylopiidae) [4]. Due to its high effectiveness, the moth was later introduced on the Caribbean Island of Nevis in 1957 to reduce densities of native and introduced cacti for the benefit of cattle holders [5]. In 1989, *C. cactorum* was detected for the first time in Florida, USA, and has since spread westward to Texas (Ken Bloem at USDA-APHIS, personal communication) and northward to North Carolina (Jarred Driscoll, North Carolina Department Agriculture and Consumer Services, personal communication). In 2006, the moth was detected on the Mexican islands of Mujeres and Contoy. As the center of origin of *Opuntia* diversity and with a large industry based on *O. ficus-indica* (L.), swift action was taken by the government to eradicate the moth through an early detection and rapid response program [6]. The control measures involved pheromone-baited traps, the sterile insect technique (SIT) [7,8], and host plant removal [9]. These measures were also used in the continental USA to slow the moth’s dispersion. However, the regional management program was canceled in 2012 due to high costs and difficulties associated with applications in swampy coastal environments.

A classical biological program to control *C. cactorum* using the parasitoid wasp *Apanteles opuntiarum* Martínez and Berta (Hymenoptera: Braconidae) has been in development since its differentiation from *A. alexanderi* Brethes [10]. The broad host range of *A. alexanderi* had previously eliminated that species as a candidate biological control agent. Native to Argentina, *A. opuntiarum* is a gregarious larval endoparasitoid and koinobiont. Laboratory studies were carried out to understand its basic biology and develop rearing protocols [11,12,13]. Field surveys conducted within the moths’ native range revealed that the host range of *A. opuntiarum* is restricted to *C. cactorum* and *C. doddi* Heinrich [14]. The results of additional evaluations of non-target species native to North America corroborated the narrow host range of *A. opuntiarum* (Benda, unpublished). Among the eight parasitoids that attack *C. cactorum*, *A. opuntiarum* showed the highest parasitism rate [11,15]. To determine the environmental suitability for introducing the parasitoid in North America, ecological niche models were developed by Pérez-De la O et al. [16]. The Florida Department of Agriculture and Consumer Services, Division of Plant Industry (FDACS–DPI) recently completed the evaluation of this parasitoid and submitted in 2023 a petition to USDA–APHIS to release it as a biological control agent of *C. cactorum*, primarily due to its high specificity.

For biological control purposes, understanding the reproductive biology of potential biocontrol agents is essential to facilitate the development of mass-rearing protocols [17,18]. Understanding the role of superparasitism in gregarious parasitoids like *A. opuntiarum* can be challenging, and the distinction between gregariousness and superparasitism can be elusive [19,20], especially when dealing with endophytic hosts. It is essential to comprehend the impact of superparasitism in biological control, as it can result in the production of smaller adults and more males in gregarious parasitoids [21]. Similarly, methods of host exposure need to be investigated to optimize parasitoid production in a laboratory colony.

Our research focuses on the reproductive biology of *A. opuntiarum* and the interaction of female wasps with their hosts. Given the gregarious condition of the parasitoid, we examined the morphology of the immature stages. We investigated the occurrence of superparasitism (depositing multiple eggs in one host) and polyembryony (single egg splitting into multiple genetically identical offspring). Additionally, we studied the impact of larval exposure and host density on parasitism rates, wasp production, and the sex ratio of offspring. This information will be valuable for mass rearing, field releases, and the effectiveness of biological control programs against *C. cactorum* in North America.

## 2. Materials and Methods

### 2.1. Study Site

Experiments were conducted at the Fundación para el Estudio de Especies Invasivas (FuEDEI), Hurlingham, Buenos Aires, Argentina, and in a quarantine facility at FDACS–DPI, Gainesville, FL, USA.

### 2.2. Insect Colonies for Morphology, Polyembryony, and Superparasitism Studies

At FuEDEI, insect rearing took place at a temperature of 25 ± 2 °C, 70% RH, and a 14:10 (L:D) photoperiod. The host colony of *C. cactorum* originated from larvae collected from an *O. ficus-indica* plantation in Santiago del Estero province, Argentina (S 27.914270°; W 67.454191°). Larvae were reared on *O. ficus-indica* cladodes in 6 L vented rectangular plastic boxes with clay pellets to absorb frass and cactus exudates. Boxes were monitored every 2–3 d, and cladodes were replaced as needed until pupation. *Cactoblastis cactorum* pupae were held individually in 20 mL cups until adulthood. Once emerged, adults were held at a 2:1 sex ratio (males:females) in 3 L containers to encourage mating. After 1–3 d, eggs were collected and emerging larvae were reared as before until they reached the third instar, at which point they were exposed to female parasitoids of *A. opuntiarum* in the studies. *Apanteles opuntiarum* adults were obtained from naturally parasitized *C. cactorum* larvae that were collected from the same *O. ficus-indica* plantation as the host larvae [15]. Wasps were identified by L. Varone and J. J. Martínez and vouchers are deposited at FuEDEI. The host larvae were reared for only one generation until pupation and/or the pupation of the wasp inside the larvae, and parasitized larvae were held individually in 20 mL cups until emergence of the adult wasps. Upon emergence, female wasps were placed in a vented 3 L clear plastic jar for 24 h with twice as many males (4–6 females with 8–12 males) to facilitate mating. A slice of *O. ficus-indica* (6 × 10 cm), a tablespoon of *C. cactorum* larval frass to stimulate female oviposition [13], and a damp strip of paper towel (1 × 3 cm) saturated with water and organic wildflower honey (“honey strip”) were added to the jar to provide food and moisture.

### 2.3. Morphology of Preimaginal Stages

The morphological development of preimaginal stages of *A. opuntiarum* was described according to Fischer et al. [22]. The cephalic structures of the mature larvae were described following Čapek [23] and Mason [24]. To observe this development, third instar *C. cactorum* host larvae were parasitized for varying periods (0, 12, 24 h, and every 24 h for 19 d) before dissection. The process was initiated by placing groups of 5–10 third instars in an 8.5 cm Petri dish with a female *A. opuntiarum* at room temperature until the female stung a larva. Once stung (usually within 10 min), each larva was removed and reared separately on a portion of *O. ficus-indica* cladode in a plastic container. A total of 62 host larvae were dissected in saline solution: 21 larvae were reared for 0–48 h to obtain 95 eggs of *A. opuntiarum,* and 41 larvae were reared for 2–19 d to obtain 105 *A. opuntiarum* larvae at different stages.

At the end of the prescribed periods, the preimaginal stages of *A. opuntiarum* were dissected from the parasitized larvae for measurement and description. The eggs were mounted on a drop of saline solution without any additional treatment. The instars were cleared with KOH and heat for 5 min at 90 °C and then mounted onto a slide to measure the head width and mandible length and describe the perioral sclerites of the mature larva. Morphological observations were documented with photographs. All measurements and photographs were taken using a Axiostar Plus microscope (Zeiss, Buenos Aires, Argentina) and a Primostar microscope with an Axiocam ERc5s camera (Zeiss). The larvae were critical-point dried and covered with a gold–palladium coating for SEM images, taken with a Philips XL30 TMP New Look microscope from the Museo Argentino de Ciencias Naturales Bernardino Rivadavia, Buenos Aires, Argentina.

### 2.4. Polyembryony and Superparasitism

To investigate developmental lifestyles in relation to polyembryony and preference for superparasitism in *A. opuntiarum,* two studies were carried out. The first study evaluated the existence of polyembryony by determining the clutch size, which represents the number of eggs laid by a female after a single oviposition event, which will be called “stung”. A mated 2 d old female wasp not previously exposed to hosts was placed with a single third instar in an 8.5 cm Petri dish and monitored until the larva was stung, at which point it was immediately removed. A total of 20 female *A. opuntiarum* wasps parasitized 20 host larvae, with 10 larvae immediately dissected after the first insertion of the ovipositor to count the number of eggs laid, and the remaining 10 reared until the emergence of the parasitoids to assess clutch size. The larvae were reared individually in 500 mL containers, and dissection and egg counting were performed using an Olympus SZ61 stereoscopic microscope (Olympus, Buenos Aires, Argentina).

In the second study, superparasitism was evaluated with a dual-choice test to determine whether female wasps prefer to oviposit in parasitized or non-parasitized larvae of different ages. Each female was exposed to two (parasitized vs. non-parasitized) third instar *C. cactorum* in an 8.5 cm Petri dish. Parasitized larvae were presented at 1, 2, or 5 d after parasitization to investigate whether the time since parasitization impacted the female wasp’s choice. The study was replicated six times. During the study, the number of times each host larva was stung and which individual was attacked first were recorded. The larvae were exposed either for 10 min or until both larvae were stung, whichever came first. Both larvae were dissected to confirm parasitism. If not hatched, the new clutch was distinguished from the previously oviposited clutch by the smaller size of the eggs (Figure 1a,b).

### 2.5. Insect Colonies for Larval Exposure and Host Density Studies

These studies were conducted at FDACS–DPI in a quarantine facility, using wasps collected in Argentina. These wasps were used to establish a colony of *A. opuntiarum* reared and studied under the following conditions: 25 ± 2 °C, 40–50% RH, and 14:10 (L:D, full spectrum light), prior to our recognition of the importance of ~70% RH. *Apanteles opuntiarum* was reared on *C. cactorum* as described by Awad et al. [12] with the following amendments to increase mating and production of female wasps: wasps were mated in larger groups using only females > 3 d old. The rearing of *A. opuntiarum* can be summarized as follows. Wasps were mated for ≥24 h by placing ≤ 100 wasps, 0–2 d old, into a vented 3 L clear plastic jar, in a ratio of 2:3 females to males, with a slice of *O. ficus-indica* (6 × 10 cm) and two honey strips. *Cactoblastis cactorum* larvae were reared on a kidney bean-based diet for ca. 3 weeks, then groups of 20 third or fourth instars were allowed to infest a 10 × 18 cm piece of cactus cladode for 24 h. The infested cactus was then placed in a vented 3 L clear plastic jar with six 1–2 d old mated parasitoids and two honey strips for 5 d. After exposure to the wasps, the infested cactus was removed from the jar and maintained in a 4 L vented plastic tub lined with a paper towel (Kimberly Clark Co., Irving, TX, USA), with frass removed and cactus replaced as needed every 2–3 d until pupation of the host or the wasps. Wasp pupae were then held individually in Petri dishes until adult emergence and the process started again. Wasp production ranged from 400–900 adult wasps weekly.

### 2.6. Host Exposure and Host Density Studies

To determine if the cactus plant material impacts host parasitization by the wasp, we evaluated the effect of larval exposure arena on parasitism rate and wasp production using a completely randomized design. *Cactoblastis cactorum* larvae were initially reared on a kidney bean-based diet for three weeks [12]. Groups of 20 third or fourth instars were allowed to infest cacti for 24 h, as described in the rearing methods. Larvae were then either exposed to six mated wasps (1–2 d old) within the cactus (“cactus-exposed”) as in the rearing methods or were removed from the cactus and placed on a 10 cm diameter Petri dish immediately prior to being placed into the jar (“Petri-exposed”). The exposure time was 5 d for larvae inside the cactus and 1 d for unprotected larvae, as the unprotected larvae had no food. After exposure to the wasps, the hosts were removed from the exposure jar and maintained in a plastic tub, as described in the rearing methods, until pupation of the host or the wasp. The numbers of parasitized, pupated, and dead hosts were recorded. Hosts unaccounted for were considered dead, since they must have decayed or been cannibalized in the days between tub maintenance. The number of wasps emerging from parasitized hosts was also recorded.

To determine the optimal number of hosts to offer wasps for maximal wasp production, we assessed the effect of presenting 10, 20, and 30 hosts on parasitism rate and wasp production. The host density experiment was conducted independently for each larval exposure arena, Petri dish and cactus. For each arena type, we implemented a randomized incomplete block design with the number of host larvae as a treatment and replicate as a random effect. Incomplete blocks were used due to the availability of wasps and hosts. *Cactoblastis cactorum* larvae were exposed as described in the larval exposure methods, except in groups of 10, 20, or 30 larvae. Each group was set to infest cacti on the same day and exposed to females from the same cohort of wasps to reduce variability in parasitoid quality within replicates. The maintenance of the exposed larvae and data collected were the same as in the larval exposure arena assays.

### 2.7. Data Analysis

#### 2.7.1. Morphology, Polyembryony, and Superparasitism

All taxonomic and development time measures in the study of the morphology of preimaginal stages are reported as mean ± SE.

To evaluate polyembryony, a two-sample *t*-test comparing the clutch size of *A. opuntiarum* after a single oviposition and after rearing was conducted (InfoStat version 2008) [25].

#### 2.7.2. Preference Testing

Thurstone’s comparative judgments model was employed to assess the detection of previous parasitization and preference for superparasitism over time. This model, utilizing the V scale [26], assigns preferences on a continuous unidimensional scale, measuring preference levels based on age since previous parasitization. The attractiveness of stimuli is determined by:(1)$$s_{i}∼\;N(\mu_{i},\;\sigma )$$
where *µ_i_* is the physiological scale value of parasitization status, *s_i_* is the realized attractiveness of stimulus i (the age since previous parasitization of exposed hosts), and *σ* is its discriminal dispersion. The model assumes homoscedasticity, and the probability of choosing one stimulus over another (P(*s_i_* > *s_j_*)) is calculated based on the joint probability distribution of the two normal distributions:(2)$$s_{i}-s_{j}∼\;N(\mu_{i}\;{-\mu }_{j},\sqrt{2\sigma })$$

The Thurstonian scale, lacking a true zero point, establishes the attractiveness of non-parasitized hosts as the origin. The scale units are defined in terms of the standard deviation (*σ*) of the stimulus distribution. Non-attacked hosts have an attractiveness of zero, and the attractiveness levels of other hosts are expressed in terms of the number of standard deviations from this baseline. The host attractiveness probability is measured with the reference point being the attractiveness of non-parasitized hosts, and all other hosts are measured relative to this standard.

The Thurstonian model parameters and their credibility intervals were calculated using a Bayesian approach with Markov Chain Monte Carlo and the Metropolis–Hastings algorithm [27]. Due to a lack of prior information on the variable distribution, a non-informative normal distribution with a mean of zero and a deviation of ten served as the a priori distribution for all parameters. Convergence was assessed using Geweke plots and visual inspection of variable traces [27,28]. The analysis was conducted using the PyMC library version 2.4 for Bayesian estimation in the Python programming language [29].

### 2.8. Larval Exposure and Host Density Studies

Data variables were calculated as follows for both the larval exposure arena and host density studies. Percent parasitized was calculated by dividing the number of parasitized hosts by the number of hosts offered in the exposure jar. Percent mortality was calculated by dividing the number of dead hosts by the number of hosts offered in the exposure jar. Mean wasp offspring per parasitized host was calculated by dividing the total wasp offspring from an exposure jar by the number of parasitized hosts collected from that jar.

Due to the large number of zeros in the number of parasitized hosts, this data set was dichotomized into two groups with either 0 or >0 parasitized hosts. A z test for difference of two independent proportions was conducted to determine if there is a significant difference in production of parasitized hosts between hosts exposed in the cactus and Petri dishes.

We used Levene’s test to confirm the homogeneity of variance and Shapiro–Wilk’s test to assess the normality of residuals for total wasp offspring and mean wasp offspring per parasitized host, which were consequently square-root transformed to improve these aspects of the data sets. The effect of larval exposure arena on the transformed data of these two variables was then analyzed using independent-sample Student’s *t*-tests. The results are presented as mean ± SE.

The data from the host density study were analyzed using a linear mixed model fit with REML, where host density was a fixed effect and replicate was treated as a random effect (jamovi Version 2.2.5). Data residuals were tested for normality using the Shapiro–Wilk test. Post-hoc *t*-tests used a Bonferroni correction for multiple comparisons. The results are presented as mean ± SE.

## 3. Results

### 3.1. Preimaginal Stages

The *A. opuntiarum* egg had a cylindrical shape with rounded ends, typical of Braconidae, Hymenoptera. The eggs were found free in the hemocoel of the host larvae. The chorion was cylindrical, transparent, and smooth, with a short tail at one end. Newly laid eggs (1 h) had a length of 309.2 ± 27.3 μm (n = 10) and width of 65.2 ± 16.4 μm (n = 10) (Figure 1a). The eggs swelled significantly over time as they probably absorbed water and nutrients from the host during development, increasing their size considerably after 24 h (Figure 1b). The eggs were found to hatch after ca. 48 h.

*Apanteles opuntiarum* has three instars characterized by the mandible (Md) morphology and length (Figure 2a–c). The development time of the first instar larva was 9.6 ± 5.2 d (n = 82), of the second instar was 13.2 ± 0.5 d (n = 7), and of the third instar was 17.3 ± 1.4 d (n = 15). The head and mandible size measurements were plotted, and three easily distinguishable groups were identified (Figure 3). The first two instars molted inside the host, while the third exited the host to spin a cocoon and pupate.

The first instar hatched as mandibulate–caudate, with the body whitish yellow, slightly wider in the anterior region, and curved to the ventral side, with 11 segments, each with a transverse line of dorsal setae, and the last segment differentiated as a short tail (Figure 1c). The first-instar head width was 274 ± 102 μm. The Md was well-developed with a slender blade without denticles (Figure 2a). The Md length was 35.6 ± 4.8 μm.

In the second instar, the body was hymenopteriform, whitish transparent, slightly broader in the anterior region, and curved to the ventral side. The head and tail dimensions were reduced in relation to the body, and the anal vesicle bilobate (Figure 1d). The Md was bidentate (Figure 2b). The head width was 668 ± 96 μm and the Md length 112 ± 6.4 μm.

In the third instar, the body was hymenopteriform and yellowish-white. The head dimensions were reduced compared to the body dimensions, and there was a reduction in the anal vesicle, with an almost imperceptible tail. The head width was 543 ± 52 μm (n = 15) and the Md length 153 ± 4.5 μm. The spiracles were well-developed. The integument was sculptured with dispersed dorsal setae (Figure 4). The third instar is the only one with functional spiracles: one pair on the mesothorax and seven pairs on abdominal segments I to VII.

The pupa was exarate, white, and enclosed in a white silk cocoon oval. The mature larval head structures were generally congruent with other microgastrine genera and the mandibular morphology typical of *Apanteles* Foerster s. s. These features include: labial sclerite broadly oval; epistoma indistinguishable; hypostoma long with a short hypostomal spur basally; stipital sclerite distinct, well developed; mandibles crossed, with the blade emerging from a large base (Figure 2c). Maxillary and labial palpi were papilliform, with two sensilla each and with a single stout seta basally. The mandibular blade had 18–20 denticles and two sharp apical teeth. The mandibles were retracted in the preoral cavity and not visible externally in the SEM images (Figure 4a). Three sensilla were located in a row on each side of the epistomal area, with two bigger sensilla directed medially and a smaller one directed laterally; exceptionally, some specimens had a fourth sensillum (Figure 2d and Figure 4a,b).

### 3.2. Polyembryony and Superparasitism

Our results showed that *A. opuntiarum* is not a polyembryonic species, i.e., each egg produced only one individual. The host larvae dissected after one oviposition event had a clutch size of 12.9 ± 4.5 eggs per host larva (range = 4–21, n = 10), which was not significantly different from the mean of 17.3 ± 7.5 adult wasps that emerged from each reared host larva (range = 7–30, n = 10) after being parasitized once (*t* = 1.7; *df* = 18; *p* = 0.1). In the preference test conducted (n = 18 females tested) between unparasitized and parasitized larvae of different ages, we observed that 11 of 16 *A. opuntiarum* females oviposited in both larvae. Two other females did not parasitize any larva within 10 min of observation. The results of the study are best described with the Thurstone scale (Table 1). The scale showed that the attractiveness of the parasitized hosts increased compared to the attractiveness of the non-parasitized host, which was used as a reference for the scale (0 d). Moreover, the attractiveness of the parasitized hosts increased during the first two days and decreased significantly on the fifth day after parasitization (Figure 5).

### 3.3. Larval Exposure and Host Density Studies

Larvae exposed in Petri dishes resulted in parasitism more frequently (z test value = 5.24, N = 88, *p* < 0.001, Table 2) and a significant increase of 1.5 times more offspring (Student’s *t* = −2.051, *df* = 55, *p* = 0.045) than those exposed in cacti (Table 3). The mean wasp offspring per parasitized host was similar in both treatments (*U* = 384, *p* = 0.993).

The larval exposure arena affected the impact of host density. When larvae were exposed in cacti, the host density did not significantly affect the percent of hosts parasitized (parasitism rate) or the number of wasp offspring produced per parasitized host (Figure 6a and Figure 7a, respectively, *p* ≥ 0.620). However, when larvae were exposed in Petri dishes, the parasitism rate and the number of wasp offspring per parasitized host were both higher when ten larvae were presented than at higher densities (Figure 6a, n = 18–23, *F*_2, 34.4_ = 6.33, *p* = 0.0005 for parasitism rate and Figure 7a, n = 18–22, *F*_2, 52.0_ = 7.64, *p* = 0.001 for wasps per host). There was no effect of host density on host mortality (Figure 6b, n = 17 to 23, *p* ≥ 0.720) or on the total number of wasp offspring produced when larvae were exposed either within cacti or in Petri dishes (Figure 7b, n = 10–22, *p* ≥ 0.414).

## 4. Discussion

Our findings indicate that the early instars of *A. opuntiarum* are similar in morphology to the Microgastrinae immature stages. However, these studies are scarce and focused mainly on solitary species [30,31,32,33,34,35]. This study provides the first description of the immature stages of a gregarious *Apanteles* species. Interestingly, the mature larval morphology of *A. opuntiarum* exhibits unique characters that have not been reported in other microgastrines [23,24]. The most distinctive character is the presence of placoid sensilla in the epistomal region, surrounded by a pigmented area. Similar to our observations, other studies have found that some endoparasitic braconids have anal vesicles during their first and second instars. Although Thorpe [36] suggested that this structure was involved in respiratory function due to its close proximity to the aorta, experiments by Edson and Vinson [37] showed that this proximity had no effect on gas exchange and was instead used for excretion. Edson and Vinson [38] and Kaeslin et al. [39] suggested that the anal vesicle also plays a critical role in nutrient intake.

The similarity in the number of eggs laid and the number of wasps emerged after a single oviposition event provides evidence that *A. opuntiarum* is not a polyembryonic species. Polyembryony is a form of clonal development that produces two or more genetically identical zygotes from a single egg, is a common mechanism in endoparasitoids of the braconid family, and has evolved independently multiple times in the Hymenoptera [40]. If *A. opuntiarum* were polyembryonic, we would have expected a larger number of emerged wasps than the number of eggs deposited. Instead, we found clutch sizes after a single oviposition similar to those of an *A. opuntiarum* laboratory colony running for six generations, where wasps were exposed to hosts within a cactus cladode [12]. On the other hand, the clutch size was smaller than those of a single-generation laboratory colony where wasps were exposed to hosts in a similar design to our Petri dish exposure arena, without shelter from a cactus cladode [11]. This finding suggests that the clutch sizes reported by Awad et al. [12] resulted from a single parasitization event and that multiple oviposition events are necessary to achieve the clutch sizes observed by Mengoni Goñalons et al. [11]. To avoid low wasp production from cactus-exposed hosts in our current FDACS-DPI laboratory colony, we compensate by providing a higher ratio of wasps to hosts (six wasps for twenty hosts).

Our results show that *A. opuntiarum* can distinguish between larvae that were parasitized vs. non-parasitized. The wasps showed an oviposition preference for 1–2 d old parasitized larvae. In this species, superparasitism is a common behavior under laboratory conditions (the current colony at FDACS–DPI typically produces 40–70 wasps per host) and is supported by larger clutch sizes in the field, with a mean of 77 cocoons per parasitized larva and 65 emerged adults. To recognize a previously parasitized host, the wasps must accomplish two steps. First, the parasitized host must be marked in some way, and second, the mark must be detected and recognized [41]. For gregarious parasitoids, adding further progeny to a host does not necessarily increase mortality, unless the host´s carrying capacity is exceeded. The decrease in attractiveness after day two in the current study may be due to limited resources for the second clutch or due to the increased ability of the older clutch to cannibalize the new arrivals [42]. The older clutch would have hatched within 48 h of being oviposited, according to our observations. Therefore, females may avoid oviposition on a host that has been parasitized for an extended period, as the suitability of the host for their offspring may be compromised.

The results from the exposure arena and host density studies also provide evidence and context for superparasitism in *A. opuntiarum*. When hosts were exposed in a Petri dish, the increase in mortality and in number of wasps per host at lower densities suggests that females parasitize individual host larvae more times if hosts are limited. Similarly, the higher mortality of host larvae in Petri dishes vs. within cacti suggests that hosts were parasitized more times when they were more accessible. A similarly inverse relationship between host availability and superparasitism has been found for other parasitoid wasp species [43,44]. The lack of change in wasp production with increased host density suggests that additional parasitization events at lower host densities resulted in successful ovipositions and also suggests that a cohort of ten hosts, parasitized multiple times, depleted the mature egg supply of the six maternal wasps in the exposure jar over the course of 24 h in a Petri dish or 5 d in cacti. This result is concordant with results from [11], where wasp production did not increase with more than 30 host larvae. The presence of mature and immature oocytes in dissected wasps [11] and ongoing studies of the effect of wasp age on reproduction suggest that this species is synovigenic. Highlighting the parasitization success of the Petri dish exposure, we measured higher parasitism in that arena despite a much shorter exposure time (1 d in a Petri dish vs. 5 d in cacti). In other species, the effect of host density on colony-reared parasitoid production can be inverse (e.g., [45]) or direct (e.g., [46]), perhaps affected by factors such as host quality, superparasitism, egg limitation, and host marking. As an example of egg supply in another braconid species, wasp production had not yet plateaued at 160 ± 8.6 progeny when two colony-reared *Habrobracon hebetor* (Say) females were exposed to 50 host larvae for 5 d in a similar host density study [46].

Superparasitism was once thought to be the result of a lack of discriminatory ability and therefore detrimental to parasitoid competitiveness [47]. However, later theories have defended the adaptive value of superparasitism and suggested that it occurs in a wide variety of hymenopteran parasitoids [48,49,50,51]. Several hypotheses have been put forth to explain why parasitoids engage in superparasitism. These include explanations related to the quality and quantity of available hosts [48]. Superparasitism may be adaptive if the wasp’s offspring has a reasonable chance of surviving the competition with the first parasitoid larva(e) and if it does not result in overly small adults [41].

The results of the exposure arena and density studies were impacted by larval defensive behavior. The higher host mortality in the Petri dishes compared to within the cacti is a straightforward demonstration of the protection gained by the larvae’s internal feeding behavior within its host plant. In addition, the larvae create and aggregate under a silk covering in both the Petri dish and the cactus. The larvae reduce their overall exposed surface area through silk and aggregation and also actively thrash and bite to avoid parasitism, similar to other gregarious herbivores [52,53]. Interestingly, Allen [53] found that groups of noctuid larvae (Lepidoptera: Noctuidae) aggregated further in response to parasitoid attack. Higher parasitism rates and numbers of wasps per host in the lowest density treatments in Petri dishes suggests that this defensive behavior is more effective in larger groups, where exposure is mostly limited to the edges of the aggregation. On the other hand, since the proportion of hosts parasitized did not increase when more than 20 hosts were exposed in Petri dishes, the number of hosts parasitized appears limited by larval defense behavior, rather than caused by a preference to superparasitize hosts. Though not significant, Mengoni Gonalons et al. [11] similarly found a decrease in the proportion parasitized by *A. opuntiarum* as the host density was increased. Variation in larval defense behavior caused by variation in aggregation and tunneling within the plant material may also explain the lack of significant differences between density treatments when hosts were presented within the cactus.

Regarding laboratory rearing optimization, our results showed that exposing host larvae to *A. opuntiarum* in a Petri dish produced more parasitized hosts and more wasps than exposing them within a cactus (Table 3). Of the densities tested, ten hosts in a Petri dish produced the most parasitized hosts and wasps per host. On the other hand, the percent parasitized within cacti did not differ between density treatments of ten and twenty hosts. Since variation was lower for 20 hosts, the optimal host density when exposing hosts within cacti is cohorts of 20 hosts.

Insofar as the behavior is subject to natural selection, exposing host larvae within the cactus may be important to maintain the natural hunting behavior of the wasps [54,55]. As a biological control agent in the field, the wasps will need to enter the cactus cladode and navigate the complexity of the tunnels, frass, and plant rot encountered therein. To take advantage of the higher wasp production rate of Petri dish exposures and still maintain selection pressure for hunting behaviors, our current laboratory protocol is to conduct only one quarter of colony exposures using larvae in Petri dishes. Because larvae are exposed in Petri dishes for less time than in cacti (1 d vs. 5 d, respectively), and since these wasps appear to be synovigenic, using the same wasps to parasitize subsequent host cohorts may be a way to increase production in a mass-reared colony. The productivity of these successive exposures needs to be assessed.

In conclusion, the first complete description, reported here, of the immature stages of a gregarious *Apanteles* species provides crucial information for the systematics and taxonomy of this species and family. An immediate example of its application is how the information on wasp instar development time lent insight into superparasitism preference. Comparative studies of closely related species, for example, the cryptic sibling species *A. alexanderi*, could be especially useful in areas of co-habitation, such as northern Argentina. Superparasitism is common in *A. opuntiarum* and preference is affected by the time elapsed since previous ovipositions. Our results also indicate that superparasitism increases with lower host availability and that parasitism is affected by host group density and behavior. Based on these results, we recommend exposing six mated wasps to cohorts of 20 host larvae and exposing host larvae in Petri dishes to maximize wasp production, but to limit Petri dish exposures due to the concern that exposing host larvae within the cactus may be important to maintain the natural hunting behavior of the wasps. This concern requires further investigation and will require multiple generations of a secondary colony to assess. The effect of host density on parasitism rate will also inform our development of biological control release protocols and release rates. Moreover, research currently underway is assessing the impact of longer-term feeding and tunneling, more reflective of field conditions, on host distribution and parasitism rates. At DPI, the rearing of *A. opuntiarum* is only being carried out under quarantine conditions, as the petition to release this parasitoid in the USA is still being evaluated. If the release is approved in the future, all the knowledge gained during the quarantine period might be transferred to a biofactory.

## Figures and Tables

**Figure 1 insects-15-00604-f001:**
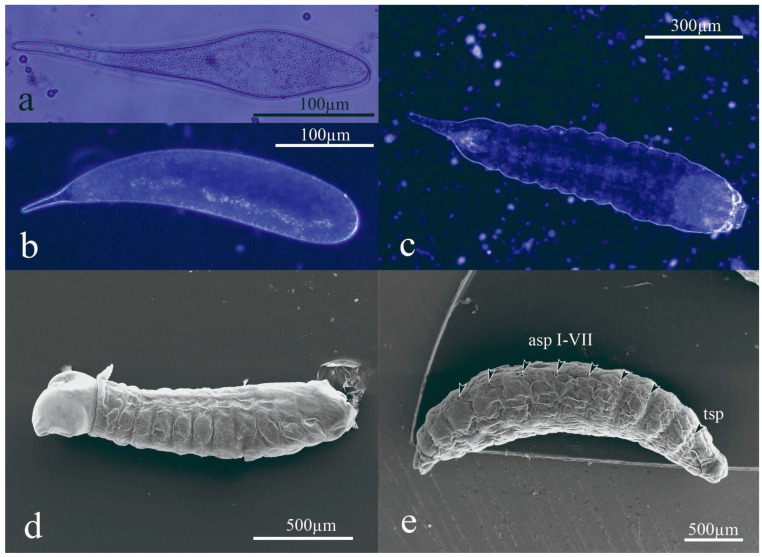
Immature stages of *Apanteles opuntiarum*: (**a**) egg 2 h after oviposition; (**b**) egg 24 h after oviposition; (**c**) first larval instar; (**d**) second larval instar with short tail; (**e**) third larval instar. References: asp I-VII., abdominal spiracles on segments I to VII; tsp., thoracic spiracle.

**Figure 2 insects-15-00604-f002:**
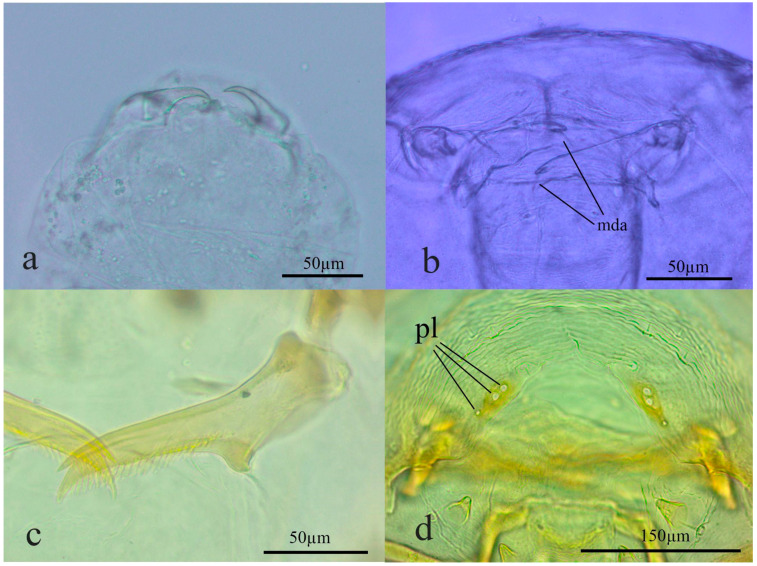
Slide mount of instars of *Apanteles opuntiarum:* (**a**) mandibles of the first instar in ventral view; (**b**) mandibles of second instar in ventral view; (**c**) mandible of third instar in ventral view; (**d**) placoid sensilla in the epistomal areas of third instar. References: mda., mandibular apex; pl., placoid sensilla.

**Figure 3 insects-15-00604-f003:**
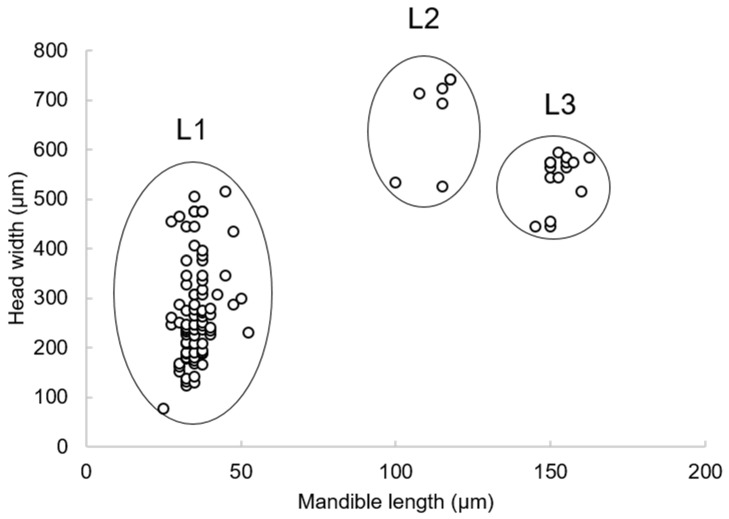
Distribution of measurements of head and mandible of larvae of *Apanteles opuntiarum,* evidencing 3 size categories, and hence, three instars (L1–L3).

**Figure 4 insects-15-00604-f004:**
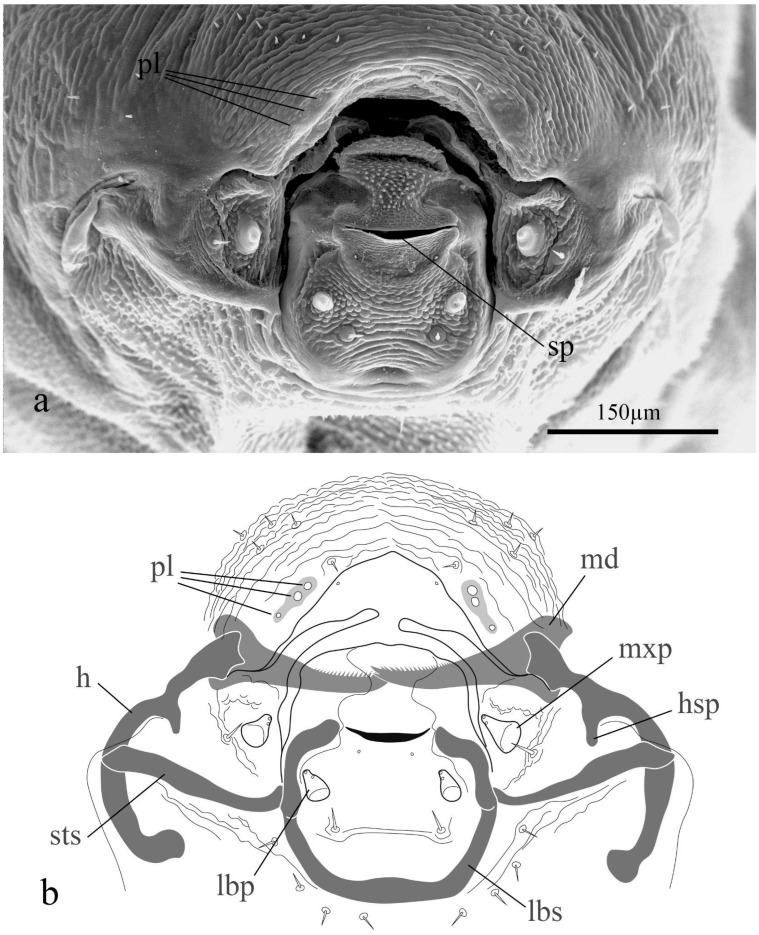
Cephalic structures of the third instar of *Apanteles opuntiarum:* (**a**) SEM image of the mouthparts; (**b**) scheme of mouthparts. References: h., hipostoma; hsp., hipostomal spur; lbp., labial palp; lbs., labial sclerite; md., mandible; pl., placoid sensilla; sp., silk press; sts., stipital sclerite; mxp., maxillary palp.

**Figure 5 insects-15-00604-f005:**
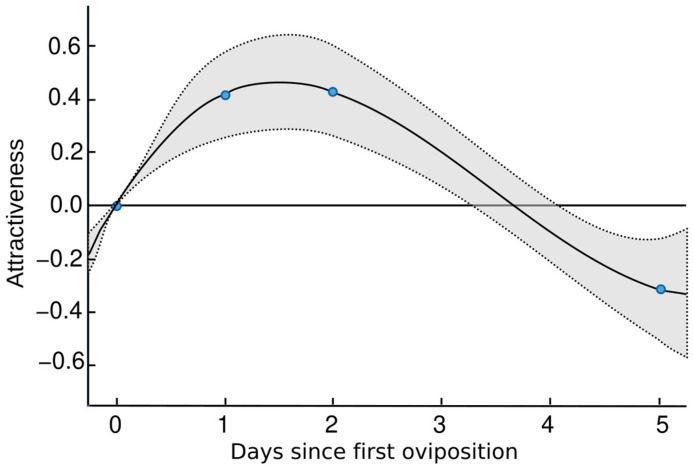
Host attractiveness to *Apanteles opuntiarum* on a Thurstone scale of attractiveness as a function of the number of days since the first oviposition by the parasitoid. The blue circles are the measurements, the solid line is the interpolated mean, and the gray area between the dotted lines are the confidence intervals.

**Figure 6 insects-15-00604-f006:**
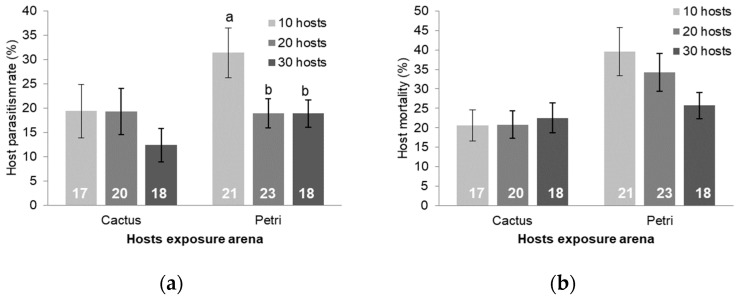
Mean parasitism rate ± SE (**a**) and mortality rate ± SE (**b**) at different host densities of *C. cactorum* exposed within cacti and in a Petri dish. Sample sizes are listed on the bars. Bars with the same letter above them are not significantly different at α ≤ 0.05 using a Bonferroni-corrected post-hoc *t*-test for multiple comparisons.

**Figure 7 insects-15-00604-f007:**
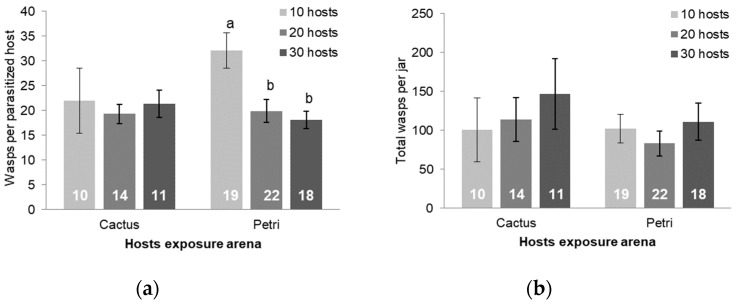
Mean number of wasps emerging per parasitized host ± SE (**a**) and total number of wasps emerging from each exposure jar ± SE (**b**) at different host densities of *C. cactorum* exposed within cacti and in a Petri dish. Sample sizes are listed directly on the bars. Bars with the same letter above them are not significantly different at α ≤ 0.05 using a Bonferroni-corrected post-hoc *t*-test for multiple comparisons.

**Table 1 insects-15-00604-t001:** Parameters of the Thurstone model on the *Apanteles opuntiarum* preference test on σ units. The attractiveness of the non-parasitized hosts was used as a reference value to create the scale and it was estimated without error. The physiological scale values of time since parasitization are represented by the character *μ*, wherein the sub-indices indicate the treatment. The non-parasitized control is labeled as 0, while 1, 2, and 5 represent the number of days since the initial oviposition.

Parameter of Time Since Parasitization	Median	2.5% CI	97.5% CI
*μ* _0_	0	0	0
*μ* _1_	0.410	0.198	0.615
*μ* _2_	0.423	0.200	0.663
*μ* _5_	−0.315	−0.575	−0.065

**Table 2 insects-15-00604-t002:** Contingency table for production of parasitized hosts when groups of host larvae were exposed to *A. opuntiarum* while within cactus or in a Petri dish.

	Parasitized Hosts Produced			Female Offspring Produced		
Exposure Arena	Yes	No	Total	Yes	No	Total
Cactus	24	24	48	5	17	22
Petri dish	0	40	40	6	29	35
Total	24	64	88	11	46	57

**Table 3 insects-15-00604-t003:** Effect of larval exposure on wasp production parameters. Values are presented as mean ± SE.

Larval Exposure Type	No. Exposure Jars	Parasitized (%)	No. Exposure Jars w/Parasitized Hosts	Total Wasp Offspring/20 Larvae	Wasp Offspring/Parasitized Host
Cactus	48	13.1 ± 2.4	22	88.3 ± 14.6	19.2 ± 2.3
Petri dish	40	33.6 ± 2.6	35	135 ± 14.4	21.2 ± 2.4

Note: “w/” = with.

## Data Availability

The data that support the findings of this study are openly available in “figshare” at: https://figshare.com/s/7b5a08da3693d1892988 (accessed on 28 December 2023).

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
