# Peer review of "Reproductive Biology and Rearing Improvements of Apanteles opuntiarum, Potential Biocontrol Agent of the Argentine Cactus Moth, Cactoblastis cactorum"

_insects, 2024, doi:10.3390/insects15080604_

Round 1

Reviewer 1 Report

Comments and Suggestions for Authors

In this research, the authors studied the morphology and some important biological features of Apanteles opuntiarum. Tests were also conducted to optimize breeding conditions, relative to host exposure and density. The work represents an important contribution to the identification of the species, understanding of its reproductive behavior, and the implementation of mass-rearing systems. This study contains a first description of the immature stages of a gregarious Apanteles species. The discussion is comprehensive and addresses the biology and the practical aspects of rearing.

However, it is important to review and adjust several aspects to improve the work and ensure that it meets the conditions for publication. There are aspects of the form that do not conform to what is required by the journal. Particularly, bibliographical references throughout the text and in the references section should be modified. Some statistical tools could be more appropriate in some of the analyses presented. It is essential to include several graphs that are mentioned in the text but are missing.

The comments and recommendations are presented in detail below:

The citations throughout the work are not in the requested style. They must be modified

Simple summary: It is recommended to review and correct the English in this section. The style is inferior to the rest of the work.

Lines 71-72: It is recommended to include a reference that supports this statement.

Line 81: "suitability for introducing the parasitoid...." change by "suitability for introducing the parasitoid in North America..."

Lines 82-83: "The Florida Department of Agriculture and Consumer Services, Division of Plant Industry (FDACS–DPI) recently completed..." When?

Figure 1: It is important to place a ruler that indicates the size scale

Lines 279-280: "percent parasitized was analyzed using independent samples non-parametric Mann-Whitney U tests": A more specific test for this type of data (which would have to be handled as proportions) is the z test for comparison of proportions between two independent samples.

Lines 284-286: Percent female offspring from each exposure had a normal distribution of residuals and homogenous variance, so the effect of larval exposure arena was analyzed using independent samples Students’ t-tests: The problem when analyzing proportions and percentages is not limited to the fact that the data may not fit a normal distribution or have heterogeneous variances. If the proportions are close to zero or 1, the confidence intervals fall outside the 0-1 range.

This occurs because the standard error of a proportion is maximum when p=0.5 and decreases as p tends to 0 or 1. This aspect is not adequately represented when assuming a normal distribution. This is why it is advisable to use a proportion comparison test.

Line 288: "Data from the host density study was analyzed" change by "Data from the host density study were analyzed"

Figure 2: For better guidance to the reader, the authors could place arrows on these micrographs pointing out the particular details to highlight.

Figure 2: It is important to place a ruler that indicates the size scale

Figure 2: Was the magnification used the same in all cases? Which was? Otherwise, it is required to place the scale on each photograph.

Figure 3: It is important to place a ruler that indicates the size scale

Line 335: Figure 2c is missing

Line 338: Figure 4a missing. Figure 4 does not show any SEM image. Figures that are described in the text are not presented.

Line 340: "Some specimens have a fourth sensillum": The authors should indicate this detail in the corresponding micrograph

Line 340: Figures 4a-b missing. Figure 4 does not show any SEM image. Figures that are described in the text are not presented.

Figure 4: Indicate in each oval the corresponding instar

Line 351: "An additional two females (n = 18 preference tests) did not parasitize any larva...": This wording is not clear. Rewrite, please

Line 368:  "and 1.5 times more offspring" change by "and a statistically significant increase of 1.5..."

Table 2: what is this "w"? Abbreviations should be placed in the footer of the table

Figure 6: Change "N-values" by "sample sizes"

Figure 6: "Bars with the same letter 392 above them are not significantly different at a = 0.05.." according to the Bonferroni test? The multiple comparisons used should be mentioned here. 

Figure 7: "Bars with the same letter 392 above them are not significantly different at a = 0.05.." according to the Bonferroni test? The multiple comparisons used should be mentioned here. 

References: The organization and format of the references do not conform to the journal's standards.

Comments are also included in the attachment

Comments on the Quality of English Language

It is recommended to review and correct the English in the Simple summary. For the rest of the work, minor editing of the English language required

Author Response

Comments and Suggestions for Authors

In this research, the authors studied the morphology and some important biological features of Apanteles opuntiarum. Tests were also conducted to optimize breeding conditions, relative to host exposure and density. The work represents an important contribution to the identification of the species, understanding of its reproductive behavior, and the implementation of mass-rearing systems. This study contains a first description of the immature stages of a gregarious Apanteles species. The discussion is comprehensive and addresses the biology and the practical aspects of rearing.

 However, it is important to review and adjust several aspects to improve the work and ensure that it meets the conditions for publication. There are aspects of the form that do not conform to what is required by the journal. Particularly, bibliographical references throughout the text and in the references section should be modified. Some statistical tools could be more appropriate in some of the analyses presented. It is essential to include several graphs that are mentioned in the text but are missing.

The comments and recommendations are presented in detail below:

  • The citations throughout the work are not in the requested style. They must be modified

Response: citations and references were modified.

  • Simple summary: It is recommended to review and correct the English in this section. The style is inferior to the rest of the work.

Response: Done.

  • Lines 71-72: The scientific name may be abbreviated after it is first cited at length.

Response: This is actually the first time it is mentioned in the introduction.

  • Lines 71-72. It is recommended to include a reference that supports this statement.

Response: The first sentences were modified for clarification (lines 77-89). We do not have a specific reference of the start of the biological control program that involved the cooperation between USDA-ARS and FuEDEI. However, the following sentences (lines 84-96) have the appropriate references to all the work was done throughout the development of the biological control program.

  • Line 81: "suitability for introducing the parasitoid...." change by "suitability for introducing the parasitoid in North America..."

Response: Done.

  • Lines 82-83: "The Florida Department of Agriculture and Consumer Services, Division of Plant Industry (FDACS–DPI) recently completed..." When?

Response: Information was added.

  • Line 183: Why were relative humidity conditions used that were so different from those of the previous tests? Is the optimal RH for the species similar to that used in this trial or in previous ones?

Response: RH conditions were different because trials were conducted in two different laboratories and in different growing chambers. Higher RH appears to improve wasp development, but this had not been recognized yet. A note was added to the text.

  • Figure 1: It is important to place a ruler that indicates the size scale.

Response: Rules were added to all the images.

  • Lines 279-280: "percent parasitized was analyzed using independent samples non-parametric Mann-Whitney U tests": A more specific test for this type of data (which would have to be handled as proportions) is the z test for comparison of proportions between two independent samples.

           Response: In response to this and the subsequent comment and due to the high number of zeros in the data, we changed the analysis of production of parasitized larvae and female offspring.  Instead of analyzing differences in percentage parasitized between the two exposure arenas, the data sets were dichotomized into two groups, with either 0 or > 0 parasitized hosts.  

  • Lines 284-286: Percent female offspring from each exposure had a normal distribution of residuals and homogenous variance, so the effect of larval exposure arena was analyzed using independent samples Students’ t-tests: The problem when analyzing proportions and percentages is not limited to the fact that the data may not fit a normal distribution or have heterogeneous variances. If the proportions are close to zero or 1, the confidence intervals fall outside the 0-1 range.

This occurs because the standard error of a proportion is maximum when p=0.5 and decreases as p tends to 0 or 1. This aspect is not adequately represented when assuming a normal distribution. This is why it is advisable to use a proportion comparison test.

Response: See response to previous comment, but also note we ultimately decided to exclude the analysis of female offspring production, for the sake of focus and brevity (mating success was not an aim of this study).

  • Line 288: "Data from the host density study was analyzed" change by "Data from the host density study were analyzed"

Response: Done.

  • Figure 2: For better guidance to the reader, the authors could place arrows on these micrographs pointing out the particular details to highlight.
  • Figure 2: It is important to place a ruler that indicates the size scale
  • Figure 2: Was the magnification used the same in all cases? Which was? Otherwise, it is required to place the scale on each photograph.
  • Figure 3: It is important to place a ruler that indicates the size scale

Response: Rules were added to figures 2 and 3, and arrows were added to figure 2 to highlight the sensilla and mandibular detail.

  • Line 335: Figure 2c is missing.

Response: Figure 2 is in line 341; it is a figure composed of 4 images. Figure 2c corresponds to “mandible of third larval instar in ventral view”.

  • Line 338: Figure 4a missing. Figure 4 does not show any SEM image. Figures that are described in the text are not presented.

 Response:  The number of the figure was incorrect in the text, the one that corresponds to the SEM image is figure 3, not 4. Figure numbers were corrected.

  • Line 340: "Some specimens have a fourth sensillum": The authors should indicate this detail in the corresponding micrograph.

Response: As the fourth sensilla is an exception, we did not present an image of this, but the other 3 sensillum were indicated in figure 2 with arrows.

  • Line 340: Figures 4a-b missing. Figure 4 does not show any SEM image. Figures that are described in the text are not presented.

Response:  The number of the figure was incorrect in the text, the one that corresponds to the SEM image is figure 3, not 4. Figure numbers were corrected.

  • Figure 4: Indicate in each oval the corresponding instar

Response: Done.

  • Line 351: "An additional two females (n = 18 preference tests) did not parasitize any larva...": This wording is not clear. Rewrite, please

Response: The sentence was rewritten.

  • Line 368:  "and 1.5 times more offspring" change by "and a statistically significant increase of 1.5..."

Response: The text was changed following the suggestion.  

  • Table 2: what is this "w"? Abbreviations should be placed in the footer of the table

Response: Done. 

  • Figure 6: Change "N-values" by "sample sizes"

Response: Done. 

  • Figure 6: "Bars with the same letter 392 above them are not significantly different at a = 0.05.." according to the Bonferroni test? The multiple comparisons used should be mentioned here. 

           Response: Done.

  • Figure 7: "Bars with the same letter 392 above them are not significantly different at a = 0.05.." according to the Bonferroni test? The multiple comparisons used should be mentioned here. 

           Response: Done.

  • References: The organization and format of the references do not conform to the journal's standards.

Response: References were modified according to the journal´s format.

Comments are also included in the attachment.

Response: All comments in the text were addressed.

Reviewer 2 Report

Comments and Suggestions for Authors

Manuscript ID: insects-2823066 entitled “Reproductive biology and rearing improvements of Apanteles opuntiarum, potential biocontrol agent of the Argentine cactus moth, Cactoblastis cactorum” by Laura Varone, Nicole Benda, Mariel Eugenia Guala, Juan José Martínez, and Octavio Bruzzone submitted to section Insect Rearing: Reserve Forces with Commercial and Ecological Values, is a well written manuscript that add to our knowledge on the reproductive performance and immature features of an important biocontrol agent. I am in favor of supporting consideration for publication in Insects MDPI pending minor revision. I added my comments to the attached PDF for authors’ revision, some of my concerns are:

·         Reference citation should follow the rules of the journal style, arrangement in numerical order, year bold and before pages.

·         Please add scale bar size on figures.

·         Please add a conclusion replacing the final summary paragraph.

·         Usually, placoid sensilla are present on adult antennae for olfactory role, can the author infer the function of these sensilla in the larval stage? Perception of host and parasiotid’s tissues, to avoid cannibalism in gregarious koinobiont endoparasitoids?

·         Missing description of functional spiracular opening in different instars, appears in third and fourth instars only?   

·         Two pages on data analysis can be reduced.

Author Response

Comments and Suggestions for Authors

Manuscript ID: insects-2823066 entitled “Reproductive biology and rearing improvements of Apanteles opuntiarum, potential biocontrol agent of the Argentine cactus moth, Cactoblastis cactorum” by Laura Varone, Nicole Benda, Mariel Eugenia Guala, Juan José Martínez, and Octavio Bruzzone submitted to section Insect Rearing: Reserve Forces with Commercial and Ecological Values, is a well written manuscript that add to our knowledge on the reproductive performance and immature features of an important biocontrol agent. I am in favor of supporting consideration for publication in Insects MDPI pending minor revision. I added my comments to the attached PDF for authors’ revision, some of my concerns are:

* Reference citation should follow the rules of the journal style, arrangement in numerical order, year bold and before pages.

Response: References were modified according to the journal´s format.

*  Please add scale bar size on figures.

Response: Scale bars were added to all figures.

  • * Please add a conclusion replacing the final summary paragraph.

Response: Done

  • * Usually, placoid sensilla are present on adult antennae for olfactory role, can the author infer the function of these sensilla in the larval stage? Perception of host and parasiotid’s tissues, to avoid cannibalism in gregarious koinobiont endoparasitoids?

Response: Since sensilla are not frequently found in larvae, we do not have enough elements or previous bibliography to infer their function.

         * Missing description of functional spiracular opening in different instars, appears in third and fourth instars only?   

Response: clarification was added (Lines 356-358).

         * Two pages on data analysis can be reduced.

Response: This section was reduced.

Additional comments on the manuscript:

Line 417: add any reference on Apanteles as polyembryonic not gregarious.

Response: Non-gregarious species do not present polyembryony because they only produce a single individual from each host.

Line 330: add color and measurements

Response: color was added. Measurements of the cocoon were not taken since the final size of the adults and therefore the size of the cocoon can vary greatly due to various factors such as the size of the host, competition among larvae, seasonal variation, etc. Generally, head width, mesoscutum width, and forewing length are better predictors of body size in wasps than any other variable. However, since our research focused on immature stages, measuring the size of adults was not involved. We added the color of the cocoon.

All other comments in the text were addressed.

Round 2

Reviewer 1 Report

Comments and Suggestions for Authors

The authors made the relevant modifications according to the initial review. In new texts, it is necessary to place scientific names in italics, which must be corrected. 

For the rest, the text meets the conditions for publication.

The attached PDF indicates the scientific names that must be formatted in italics

Author Response

Changes were made upon request. Thank you.
